# Ligand-induced IFNGR1 down-regulation calibrates myeloid cell IFNγ responsiveness

William J Crisler, Emily M Eshleman, Laurel L Lenz

**The type II IFN (IFNγ) enhances antimicrobial activity yet also drives expression of genes that amplify inflammatory responses. Hence, excessive IFNγ stimulation can be pathogenic. Here, we describe a previously unappreciated mechanism whereby IFNγ itself dampens myeloid cell activation. Staining of monocytes from *Listeria monocytogenes*–infected mice provided evidence of type I IFN–independent reductions in IFNGR1. IFNγ was subsequently found to reduce surface IFNGR1 on cultured murine myeloid cells and human CD14+ peripheral blood mononuclear cells. IFNγ-driven reductions in IFNGR1 were not explained by ligand-induced receptor internalization. Rather, IFNγ reduced macrophage *Ifngr1* transcription by altering chromatin structure at putative *Ifngr1* enhancer sites. This is a distinct mechanism from that used by type I IFNs. Ligand-induced reductions in IFNGR1 altered myeloid cell sensitivity to IFNγ, blunting activation of STAT1 and 3. Our data, thus, reveal a mechanism by which IFNGR1 abundance and myeloid cell sensitivity to IFNγ can be modulated in the absence of type I IFNs. Multiple mechanisms, thus, exist to calibrate macrophage IFNGR1 abundance, likely permitting the fine tuning of macrophage activation and inflammation.**

## Introduction

When microbes penetrate epithelial barriers, host pattern recognition receptors detect microbial or damage-associated host products (PAMPS or DAMPs). Pattern recognition receptor ligation signals the production of cytokines and other factors important for eliciting, shaping, and amplifying inflammatory responses [1, 2]. In many cases, microbes are cleared by an initial wave of phagocytes and these inflammatory responses resolve. Persistence of inflammatory responses is associated with chronic conditions such as atherosclerosis, Alzheimer's, and cancer. To better understand and treat these diseases, there is need for an improved understanding of endogenous processes that limit and promote the resolution of inflammatory responses.

One family of cytokines critical for mediating inflammatory responses is the IFNs. Type II IFN (IFNγ) is a proinflammatory cytokine that boosts the antimicrobial functions of myeloid cells. IFNγ ligates a heterodimeric cell surface receptor, the interferon gamma receptor (IFNGR), comprising ligand-binding IFNGR1 and signal-transducing IFNGR2 proteins [3]. Ligation of the IFNGR propagates a signaling cascade involving the Janus tyrosine kinases (JAKs) JAK1 and JAK2. The activated JAKs phosphorylate tyrosine residues in the IFNGR cytoplasmic domain to stimulate recruitment of signal transducer and activator of transcription (STAT) proteins, including STATs 1 and 3. Phosphorylation of STAT1 on Tyrosine 701 (pSTAT1Y$^{701}$) induces the formation of canonical pSTAT1 homodimers, which translocate to the nucleus where they bind DNA to promote expression of IFNγ-activated genes (GAGs) [4]. Many GAGs encode proteins that boost inflammatory responses or increase myeloid cell antimicrobial activities. IFNγ stimulation of myeloid cells, thus, plays a critical role in mediating host resistance to infections by numerous intracellular bacteria and parasites [5, 6, 7, 8]. Accordingly, defects in the IFNγ response increase susceptibility to diverse pathogens, including *L. monocytogenes* (Lm) and *Mycobacterium tuberculosis* (Mtb) [9, 10, 11, 12].

The type I IFNs comprise IFN$\beta$ and at least 13 other IFN subtypes—all of which ligate the interferon alpha receptor (IFNAR) to elicit cellular responses [13]. Abundant production of these cytokines occurs and has been shown to substantially increase host susceptibility during systemic infections by Lm as well as mucosal infections by Mtb and several other bacterial pathogens [5, 14, 15, 16, 17, 18, 19, 20]. These detrimental effects of type I IFNs correlate with their ability to impair myeloid cell responsiveness to IFNγ [5, 18, 19, 20, 21]. In murine myeloid cells, reduced IFNγ responsiveness correlates with rapid silencing of de novo transcription of the *Ifngr1* gene and a subsequent decrease in surface expression of IFNGR1 [22]. Reductions in surface IFNGR1 have also been observed on CD14+ monocytes from human patients with untreated Mtb [23]. Reductions in myeloid cell surface IFNGR1 are associated with silencing of *Ifngr1* transcription due to recruitment of a repressive early growth response factor 3 (EGR3) transcriptional complex to the proximal murine *Ifngr1* promoter [24]. Our laboratory recently developed a mouse model in which this repression is circumvented because of transgenic expression of a functional flag-tagged

Department of Immunology & Microbiology, University of Colorado School of Medicine, Aurora, CO, USA

Correspondence: laurel.lenz@CUanschutz.edu

IFNGR1 (fGR1) expressed from a macrophage-specific promoter (18). Macrophages from fGR1 transgenic mice maintain IFNGR1 expression despite IFNAR ligation. The increased activation of these fGR1 macrophages by IFNγ correlates with increased resistance of fGR1 mice to systemic Lm infection (18). Together, these findings suggest that type I IFN-driven susceptibility to bacterial infections is at least partly due to reductions in myeloid cell IFNγ responsiveness.

In the present studies, we noted type I IFN-independent reductions in myeloid cell surface IFNGR1 staining in the context of systemic Lm infection. Further mechanistic investigations demonstrated that this reduced IFNGR1 was driven by IFNγ itself and associated with silencing of *Ifngr1* transcription. The mechanism for *Ifngr1* silencing was found to be distinct from that used by type I IFNs and involved altered chromatin at putative *Ifngr1* enhancer sites. The IFNγ-driven reductions in IFNGR were also associated with a transient dampening of myeloid cell responsiveness to IFNγ. These data demonstrate that there are multiple mechanisms by which IFNGR1 availability is regulated in myeloid cells, which is counter to the accepted dogma that IFNGR is constitutively expressed. Furthermore, the finding that IFNγ acts to impair accessibility of its own receptor suggests this mechanism may be important for increasing the threshold of cytokine required for myeloid cell activation; possibly helping to restrict macrophage activation and inflammatory responses when microbial translocation or other stimuli elicit limited or transient IFNγ production.

# Results

### *L. monocytogenes* infection triggers type I IFN–independent reductions in myeloid cell surface IFNGR1

Our prior studies revealed that type I IFNs drive reductions in myeloid cell surface IFNGR1, correlating with increased susceptibility to i.v. infection with Lm (18, 22). We, therefore, investigated the ability of an antibody that blocks ligation of the type I IFN receptor (αIFNAR1) to prevent reductions in myeloid cell surface IFNGR1 during Lm infection of C57BL/6 (WT) mice. Groups of mice were treated with αIFNAR1 or, as a control, an antibody that neutralizes IFNγ (αIFNγ). A separate set of mice received both antibodies before infection with $10^4$ live Lm. Splenocytes harvested at 0, 24, or 72 hours postinfection (hpi) were stained for surface IFNGR1. Surface IFNGR1 geometric mean fluorescence intensity (gMFI) was reduced by nearly 50% on gated splenic monocytes (CD11b+Ly6C+Ly6G−) at 72 hpi from the infected control (PBS) mice versus naïve controls (Fig 1A). Similar results were obtained in IFNγ-depleted (αIFNγ) mice, whereas no reduction in monocyte IFNGR staining was seen at 72 hpi in mice pretreated with αIFNAR1 or αIFNAR1 plus αIFNγ (Fig 1A). These data are consistent with prior work using B6.*Ifnar1*$^{−/−}$ mice and indicate that blockade of type I IFNs (but not IFNγ) suffices to prevent myeloid cell IFNGR down-regulation seen at 72 hpi (22).

Strikingly different results were obtained when we evaluated cell surface IFNGR1 staining on splenic monocytes at 24 hpi (Fig 1B). Here, the reductions in IFNGR1 staining were no longer blocked in the mice treated only with αIFNAR1. Neither were the reductions in IFNGR1 staining blocked in mice treated with αIFNγ alone. However,

when both αIFNAR1 and αIFNγ were given to the infected mice, myeloid cell surface IFNGR was comparable with that seen in uninfected mice. A similar pattern was seen when IFNGR1 was evaluated on gated splenic DCs (Fig S1A and B). These data indicate that during i.v. Lm infection, cell surface IFNGR1 abundance on these myeloid cells is regulated by both type I IFN–dependent and type I IFN-independent processes. Furthermore, they implicate IFNγ itself as a contributor to the type I IFN-independent IFNGR1 down-regulation seen during the first 24 h of systemic Lm infection.

In the context of the Lm infection, the early (24 h) type I and II IFN–dependent IFNGR1 down-regulation was observed to be consistent with serum IFNγ concentrations, which were elevated at 24 hpi but returned nearly to baseline by 72 hpi (Fig 1C). This early IFNγ has been shown to derive from antigen-independent T lymphocyte and NK cells responses to Lm infection (25). The abundance of *Ifng* transcripts in spleens of the control and infected mice followed similar kinetics (Fig 1D). Conversely, transcript abundance for *Ifnb* and *Ifna* subtypes were increased at 24 h and remained elevated at 72 hpi (Fig 1E and F). These increases in type I IFN production parallel the increases in bacterial burdens during the i.v. Lm infection (25). Thus, the data argue that the transient spike in IFNγ production at 24 hpi contributes together with type I IFNs to the reductions in myeloid surface IFNGR observed at this time point, with later reductions in IFNGR1 being driven solely by the type I IFNs.

### IFNγ suffices to stimulate reductions in myeloid cell IFNGR1

To investigate whether IFNγ directly impacts myeloid cell surface IFNGR1 abundance, splenocytes from C57Bl/6 mice were cultured 8 h in vitro with 100 U/ml recombinant IFNβ, IFNγ, or a combination of both. Treatments with either cytokine alone or in combination sufficed to significantly reduce surface IFNGR1 abundance on gated splenic monocytes (Fig 2A). To test if this effect was unique to murine cells, we next treated PBMCs or THP-1 monocytes with human cytokines and evaluated cell surface IFNGR1 on gated CD14+ cells (Figs 2B and C, and S2A). Surface IFNGR1 was also reduced in the treated human myeloid cells. Thus, despite divergence of the human and mouse IFNγ and IFNGR1 proteins, IFNγ treatment sufficed to reduce IFNGR1 abundance on myeloid cells from both species.

IFNγ treatment also significantly reduced surface IFNGR1 staining on BMDMs and BMDCs from WT or B6.*Ifnar1*$^{−/−}$ mice (Figs 2D and S2B), indicating the effects of IFNγ did not require type I IFN signaling. Comparison of staining of BMDMs from WT mice and those lacking IFNGR1 (*Ifngr1*$^{−/−}$) further confirmed specificity of the IFNGR1 staining (Fig S2C). However, IFNγ stimulation of WT splenocytes did not reduce IFNGR1 staining on T cells (Fig S2D). Interestingly, IFNγ stimulation also failed to significantly reduce surface IFNGR2 staining on BMDMs (Fig S2E), arguing the ligated receptor complex was not simply internalized. We further observed that treatments with either IL-6 or IL-10 were unable to reduce IFNGR1 nor IFNGR2 staining at concentrations that sufficed to induce STAT phosphorylation (Fig S2E and F). These data reveal that IFNγ is sufficient to reduce myeloid cell surface availability of IFNGR1 and that this response is not universally induced by other pro- or anti-inflammatory cytokines.

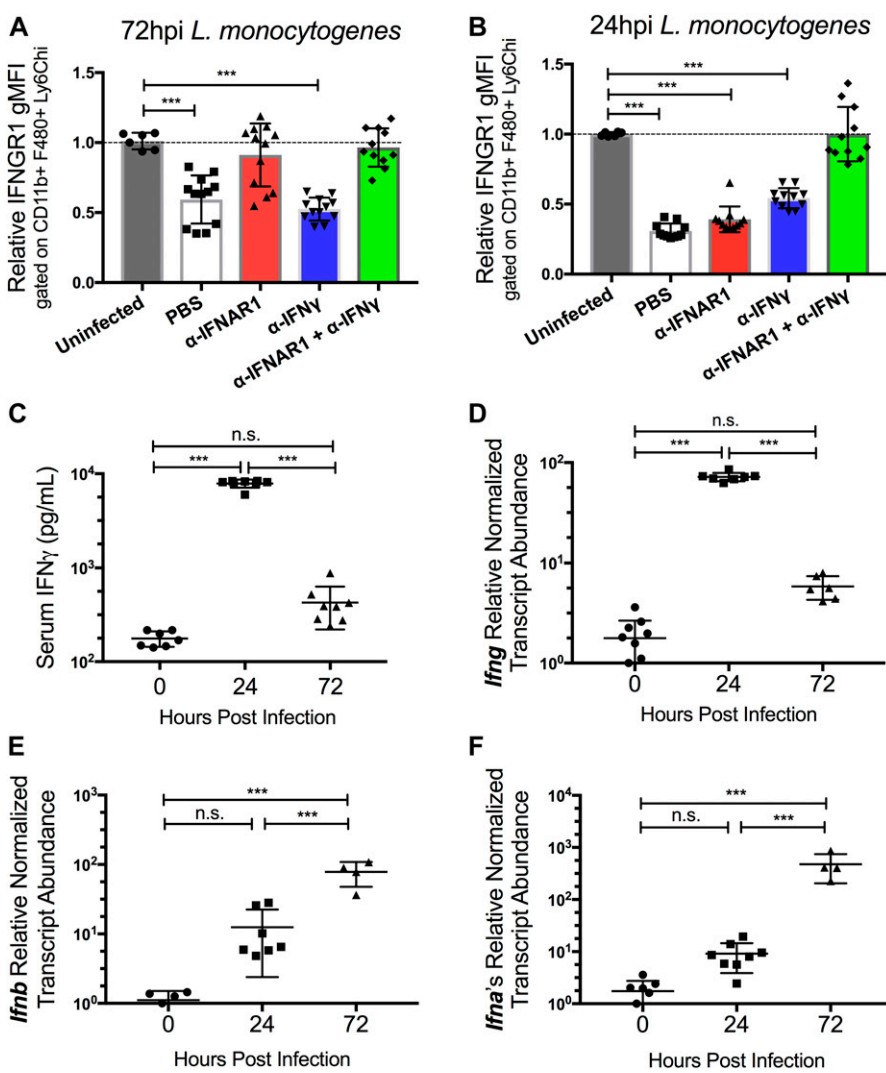

**Figure 1. Infection triggers type I IFN–independent reductions in myeloid cell surface IFNGR1 that are dependent on type II IFN (IFNγ).**
WT C57BL/6 mice were injected *i.p.* with 0.5 mg of α-IFNγ, α-IFNAR1, both antibodies, or PBS vehicle control 24 h before *i.v.* infection with $10^4$ CFU *L. monocytogenes*. **(A, B)** At 0, 24, or 72 hpi, splenocytes were harvested for FACS analysis. Splenic monocytes were gated as detailed in the Materials and Methods section. Relative changes in gMFI for IFNGR1 staining is shown versus control staining for three pooled experiments at each time point. **(C)** Serum IFNγ concentrations at the indicated times after infection. Data are pooled from three experiments with a total of 7–9 mice/group. **(D, E, F)** Relative normalized transcript abundance of *Ifng*, *Ifnb*, or *Ifna* subtypes from lysed whole splenocytes at indicated times after infection. Data are pooled from three experiments with a total of 4–9 mice/group. **(A, B, C, D, E, F)** For all panels, error bars represent SEM; ***$P <$ 0.001 as determined by one-way ANOVA and Dunnett's (A, B) or Tukey's (C, D, E, F) post-hoc tests for comparison between uninfected and other groups or comparison between conditions. n.s., not significant.

## Reduced IFNGR1 staining requires JAK signaling and does not correlate with induction of myeloid cell death

Time course experiments further revealed that reductions in IFNGR1 did not occur immediately following IFNγ stimulation, but in WT, BMDMs required at least 2 h of stimulation and continued to decline for at least 8 h (Fig S2G). These kinetics suggested that signaling through the IFNGR drives the observed reductions in surface IFNGR1 protein. To test this further, we treated cells with ruxolitinib (Ruxo), an ATP-competitive inhibitor primarily targeting JAK1 and JAK2 (26, 27). Ruxo pretreatment of BMDMs blocked the ability of IFNγ to induce down-regulation of IFNGR1 (Fig S2H). These results confirm a requirement for signaling downstream of the IFNGR and further demonstrate that ligand binding to IFNGR is not sufficient to cause the observed reduction in staining of cell surface IFNGR1. Hence, reduced IFNGR1 staining was not due to blockade of antibody binding by the IFNγ ligand.

Given the ability of type I and II IFNs to induce apoptosis in susceptible cells (28, 29), we considered the possibility that induction of cell death might contribute to reductions in IFNGR1

staining. To evaluate this, BMDMs were treated with a fixable live/dead stain before surface staining. After excluding doublets and debris, BMDMs were gated on live cells. After 8 h in culture, ~88% of untreated BMDMs were viable (Fig S2I). Viability of the BMDMs was similarly high in the cultures treated 8 h with 100 U/ml IFNβ or IFNγ alone (Fig S2I). Only when IFNβ and IFNγ treatments were combined was there a reduction in cell viability (~78% viable). The percentage of live cells was reduced further with increased concentrations of the IFNs but remained >60% even with cytokine concentrations of 1,000 U/ml (Fig S2I). By comparison, only ~16% of heat-killed BMDMs stained within the live gate (Fig S2I). When IFN-treated BMDMs from these cultures were gated on the live-stained population, reductions in IFNGR1 staining were similar regardless of the presence of dead cells in the cultures (Fig S2J). Thus, we conclude that reduced IFNGR1 staining is not an artifact of cell death induction by the IFNs.

## IFNγ stimulation reduces total cellular abundance of IFNGR1

Ligation of various cell surface transmembrane receptors is known to trigger receptor-ligand endocytosis (30), and prior studies provided

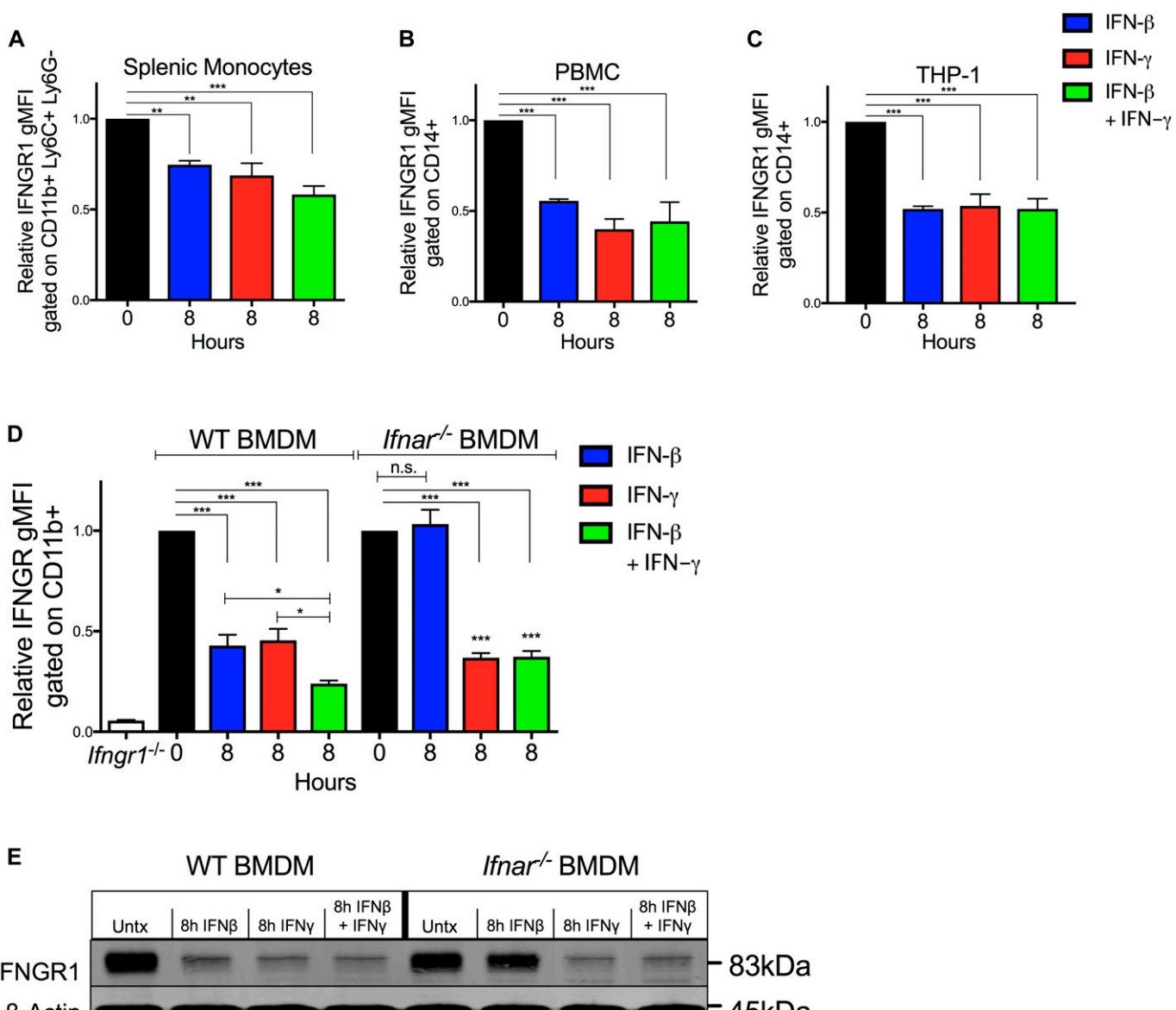

**Figure 2. IFNγ stimulation reduces surface IFNGR1 abundance on murine and human myeloid cells.**
**(A, B, C, D)** Indicated myeloid cells were treated in culture for 8 h with 100 U/ml recombinant IFNβ (blue), IFNγ (red), or IFNβ and IFNγ together (green) then stained and analyzed for cell surface IFNGR1. **(A, B, C, D)** Shown is normalized surface IFNGR1 gMFI on gated (A) CD11b+, Ly6C+, Ly6G– splenic monocytes from naïve WT mice, (B) CD14⁺ human PBMC, (C) CD14⁺ human THP-1 cells, and (D) CD11b⁺ BMDMs from WT or B6.*Ifnar1*⁻/⁻ mice. Each bar graph represents the mean ± standard deviation (SD) of the pooled values for each condition (n = 3 independent experiments). Error bars represent SEM; n.s., *P < 0.05, **P < 0.01, ***P < 0.001 by one-way ANOVA and Dunnett's or Tukey's post-hoc test for comparison between untreated ("0 h") and other groups or comparison between multiple conditions. **(E)** WT or *Ifnar1*⁻/⁻ BMDMs were treated ±8 h 100 U/ml recombinant IFNβ, IFNγ, or IFNβ and IFNγ together. Representative immunoblot of lysates probed for IFNGR1 and β-actin as loading control. n.s., not significant.

evidence for ligand-induced endocytosis of IFNGR1 in epithelial cells (31). Older studies using radio-labeled IFNγ also observed that ligation of IFNGR in macrophages transiently reduced the number of cell surface cytokine binding sites (32), consistent with either endocytosis or reduced expression of IFNGR. We, thus, further considered whether IFNγ-stimulated reductions in myeloid cell surface IFNGR1 might solely be due to ligand-induced internalization of IFNGR1 protein by evaluating the effects of IFNβ or IFNγ stimulation on total cellular IFNGR1. BMDMs from WT or *Ifnar1*⁻/⁻ mice were lysed 8 h after treatment

with 100 U/ml IFNβ, IFNγ, or IFNβ plus IFNγ. Total cellular IFNGR1 protein was quantified by immunoblot analysis. Compared with untreated WT cells, total IFNGR1 protein was reduced by >50% after 8 h treatment in each stimulation (Figs 2E and S3A). Total IFNGR1 protein was also reduced in *Ifnar1*⁻/⁻ BMDMs, but only in response to the IFNγ treatment (Figs 2E and S3A). These data indicate that the loss of surface IFNGR1 in IFNγ-stimulated macrophages is not simply due to transient receptor internalization but is instead associated with reduced cellular abundance of this protein.

## IFNγ reduces *Ifngr1* transcript abundance via a distinct mechanism from type I IFN

Given that there were reductions in total cellular IFNGR1 after IFNγ treatment and type I IFNs are known to silence de novo transcription of the *Ifngr1* gene (22, 24), we used quantitative real-time PCR (qRT-PCR) to evaluate abundance of *Ifngr1* transcripts at 0–8 h after IFNβ or IFNγ stimulation. The results indicated that treatment with either IFN type significantly reduced *Ifngr1* transcript abundance (Fig 3A). Interestingly, the reduction in *Ifngr1* transcript was more rapid after IFNβ (50% reduction at ~3.8 h) versus IFNγ (50% reduction at ~6 h). The delayed effects of IFNγ likely reflect the attenuated myeloid cell response to IFNγ in the presence of endogenous type I IFNs (5) because IFNγ treatment reduced *Ifngr1* transcript abundance much more rapidly (50% reduction at ~2 h) in *Ifnar1*$^{-/-}$ BMDMs (Fig 3B). IFNγ stimulation did not reduce *Ifngr2* transcript abundance (Fig S3B), indicating a specific effect on *Ifngr1* by IFNγ. Treatments with IL-6 or IL-10 failed to reduce *Ifngr1* or *Ifngr2* transcript abundance (Fig S3B). Thus, similar to type I IFNs, stimulation with IFNγ acts to rapidly reduce myeloid cell *Ifngr1* mRNA abundance.

To further investigate the mechanisms for reduced *Ifngr1* transcript abundance, we performed chromatin immunoprecipitation (ChIP) experiments in the control and treated WT BMDMs. Phosphorylation of RNA pol II at serine five residues within the C-terminal domain heptapeptide repeat is required for transcription initiation (33). We, thus, used ChIP to assess recruitment of pS5-RNA pol II to the *Ifngr1* promoter. Compared with control cells where *Ifngr1* is actively transcribed, pS5-RNA pol II occupancy at the *Ifngr1* transcriptional start site was significantly reduced by IFNγ treatment (Fig 3C). This result suggested IFNγ acts to block de novo *Ifngr1* transcription. Consistent with this, treatment of WT BMDMs with IFNγ reduced *Ifngr1* transcript abundance as rapidly and effectively as treatment with the RNA pol II inhibitor actinomycin D (Fig S3C). IFNβ treatment likewise blocks de novo transcription of *Ifngr1*, in this case by promoting recruitment of a repressive Egr3/Nab1 complex to a site in the proximal *ifngr1* promoter (24). We, thus, evaluated the effects of IFNγ treatment on luciferase production by RAW264.7 macrophage reporter cells (IFNGR1pr-luc) stably transfected with a proximal *ifngr1* promoter-luciferase construct that includes the Egr binding site (24). Unlike IFNβ, IFNγ treatment failed to suppress luciferase reporter activity in these IFNGR1pr-luc cells (Fig 3D). Consistent with this result, ChIP experiments revealed that IFNγ treatment of BMDMs did not enrich Egr3 protein at the *Ifngr1* promoter in WT BMDMs (Fig 3E). These results indicate that although type I and II IFNs both silence *Ifngr1* transcription in myeloid cells, they do so via distinct mechanisms.

To further define how IFNγ treatment blocks *Ifngr1* transcription, we evaluated occupancy of enhancer regions upstream of the *Ifngr1* locus that were identified based on analysis of a published genome-wide H3K4me3 ChIP-seq analysis in human monocytes (34). H3K4me3 accumulation at enhancer sites is associated with active transcription (35). Chromatin was isolated from control, IFNβ, or IFNγ-treated THP-1 human monocytes and ChIP performed with an anti-H3K4me3 antibody (Fig 3F). These experiments confirmed association of H3K4me3 at all four tested enhancer regions in control BMDMs and revealed that treatment with IFNγ (but not IFNβ)

selectively and significantly reduced H3K4me3 occupancy at two of these regions (one and three in Fig 3F). Neither treatment affected H3K4me3 association at regions 2 or 4 (Fig 3F). These findings together indicate that the rapid silencing of de novo *Ifngr1* transcription in IFNγ-treated cells is uniquely associated with changes in occupancy or accessibility of enhancer elements upstream of the *Ifngr1* gene.

The data above suggested that IFNγ suppresses myeloid cell IFNGR1 abundance by blocking de novo transcription of *Ifngr1*. To further test this, we investigated the ability of IFNγ treatment to reduce IFNGR1 staining on splenic monocytes from fGR1 mice, whose macrophages express a transgenic *Ifngr1* gene driven by the *c-fms* promoter (18). IFNγ treatment of splenic monocytes from the fGR1 mice did not significantly reduce IFNGR1 staining (Fig 3G). These data suggest that the transgenic *Ifngr1* resists silencing by IFNγ and, thus, prevents IFNγ from reducing surface IFNGR1.

## IFNγ-stimulated myeloid cells become refractory to STAT activation in response to secondary IFNγ exposure

We next considered whether ligand-induced reductions in IFNGR1 sufficed to limit macrophage responsiveness to subsequent IFNγ stimulation. Immunoblotting was first used to quantify the duration of pSTAT1Y$^{701}$ induction after a 30-min pulse of IFNγ. WT BMDMs were treated for 30 min with 100 U/ml IFNγ, washed with PBS, and allowed to rest in cytokine-free media before lysis and immunoblotting. This IFNγ pulse rapidly induced pSTAT1Y$^{701}$ accumulation in the BMDMs with the abundance of pSTAT1Y$^{701}$ decaying to nearly background levels at 5 h after stimulation (Fig 4A). We, thus, chose to re-expose BMDMs to a secondary IFNγ "hit" (100 U/ml for 5, 10, 30, or 60 min) 5 h after an initial IFNγ pulse. Strikingly, the secondary IFNγ treatment failed to induce pSTAT1Y$^{701}$ (Figs 4B and S4A). By contrast, when IFNβ was used in the secondary hit, pSTAT1Y$^{701}$ was potently induced (Figs 4C and S4B). These data indicate there was continued availability of STAT1 in the IFNγ-pulsed cells and susceptibility of this STAT1 to tyrosine phosphorylation. Induction of pSTAT3Y$^{705}$ was also abrogated in IFNγ-pulsed BMDMs upon secondary "hit" with IFNγ but not IFNβ (Fig S4C). Thus, the IFNγ-primed cells became refractory to subsequent IFNγ. This refractoriness of WT BMDMs to IFNγ correlated with reduced intensity of cell surface IFNGR1, as measured by flow cytometry (Fig 4D). Because surface staining for IFNGR1 was restored by 12 h of rest after the initial IFNγ pulse (Fig S2G), we further evaluated the ability of IFNγ to induce pSTAT1Y$^{701}$ at this later time point. The responsiveness to IFNγ was restored (Fig 4D). In contrast to WT peritoneal macrophages (Fig S4D), IFNγ-pulsed peritoneal macrophages isolated from fGR1 mice still induced pSTAT1Y$^{701}$ upon the secondary hit with IFNγ (Fig 4E), consistent with their ability to retain surface IFNGR1 (Fig 3G). These data together suggested that an initial encounter with IFNγ renders WT macrophages unresponsive to IFNγ or raises their threshold for responsiveness to this cytokine. To distinguish between these possibilities, we evaluated whether increasing the IFNγ concentration used in the secondary stimulation facilitated the induction of pSTAT1Y$^{701}$ or pSTAT3Y$^{705}$. These experiments showed that after an initial IFNγ pulse, the amount of IFNγ required for a comparable pSTAT1Y$^{701}$ or pSTAT3Y$^{705}$ response 5 h later was increased by threefold (Figs 4F and S4E). We interpret these data to indicate that

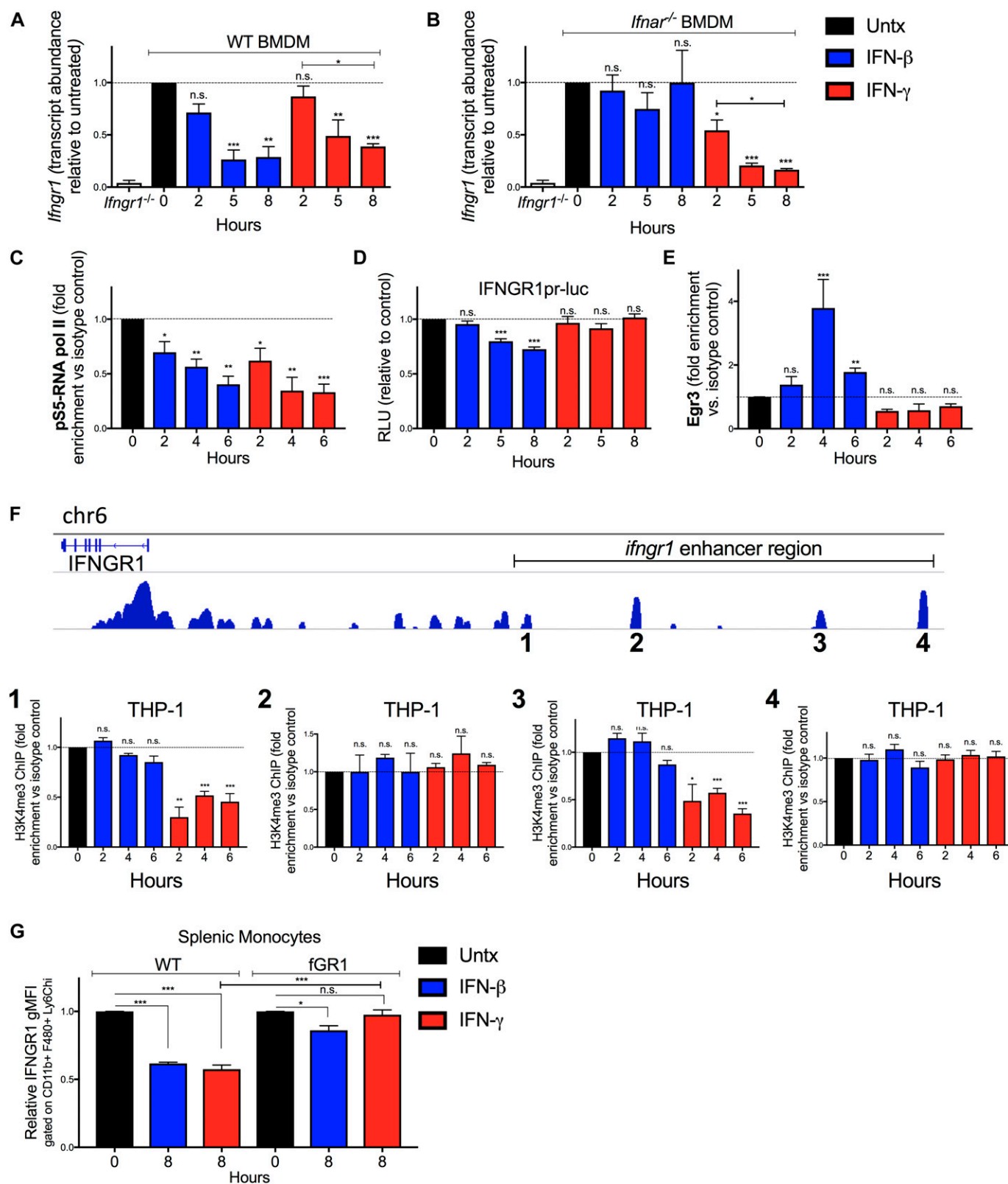

**Figure 3. IFNγ and IFNβ reduce *Ifngr1* transcript abundance via distinct mechanisms.**
WT or *Ifnar1*$^{-/-}$ BMDMs or RAW 264.7 reporter cells were treated 0–8 h with 100 U/ml recombinant IFNβ (blue) or IFNγ (red). **(A, B)** Total RNA was isolated and analyzed by qRT-PCR. The relative transcript abundance of *Ifngr1* was calculated using the $2^{-\Delta\Delta Ct}$ model with Gapdh and Hmbs mRNA as housekeeping genes. **(C)** ChIP was performed for pS5-RNA pol II in WT BMDMs. Murine primers that amplify 100 base pairs within exon 1 of *Ifngr1* were used to quantify immunoprecipitated chromatin by qRT-PCR.

IFNγ-induced reductions in myeloid cell surface IFNGR1 act to selectively and transiently dampen myeloid cell sensitivity to subsequent IFNγ.

# Discussion

More than 35 years ago, IFNγ was demonstrated to play a vital role in the induction of microbicidal and tumoricidal macrophage activation (36, 37). However, the complex processes governing macrophage activation remain incompletely defined. Excessive IFNγ responses contribute to destruction of host tissue and are associated with a variety of inflammatory diseases (38, 39, 40, 41). IFNγ signaling in myeloid cells also broadly impacts hematopoiesis (42). Mechanisms that dampen IFNγ responses are, thus, essential for homeostasis. Previously described mechanisms that attenuate cellular responses to IFNγ include posttranslational ubiquitination of signal mediators and GAGs, epigenetic changes, and induction of suppressive long noncoding RNAs (43, 44, 45, 46). Our results here have identified an additional, previously unappreciated mechanism whereby IFNγ itself acts to tune myeloid cell activation through suppressing *Ifngr1* transcript abundance and surface IFNGR1. This rapid suppression of *Ifngr1* is associated with altered occupancy of *Ifngr1* enhancer elements and drives a transient reduction in IFNGR1 surface availability that renders macrophages refractory to a second exposure with similar or lesser amounts of IFNγ. These findings add a new layer of complexity to our understanding of the IFNγ system and the regulation of IFNγ-driven myeloid cell activation.

There are similarities between the IFNγ-driven suppression of IFNGR1 described here and results from our previous work showing that type I IFNs suppress myeloid cell responsiveness to IFNγ through suppression of IFNGR1 (22, 24). However, there are also important mechanistic differences. Specifically, the silencing of *Ifngr1* transcription by IFNγ does not involve the recruitment of repressive Egr3 complexes to the proximal *Ifngr1* promoter. The existence of two distinct mechanisms for silencing *Ifngr1* by type I versus type II IFNs enables finer control of receptor expression during inflammatory responses and seemingly facilitates an increased level of IFNGR1 suppression when the two mechanisms are combined. Although not explored here, the combined suppression of IFNGR1 by type I and II IFNs may render macrophages refractory to higher concentrations of IFNγ for a longer time period. This could explain why both mechanisms of suppression are needed and have been conserved throughout the divergent evolution of the IFN

systems in mice and humans. These areas will be important for study in future work.

Our previous work showed that silencing of myeloid cell *Ifngr1* expression correlates with the detrimental impact of type I IFNs on the host's ability to resist systemic Lm infection (5, 22, 24). Type I IFNs have also been shown to increase the susceptibility to diverse additional systemic and mucosal bacterial infections (5). A transgenic strategy to circumvent this IFNGR1 down-regulation in myeloid cells was, thus, pursued and shown to boost monocyte activation and improve host resistance to systemic Lm (18). Although we initially attributed this protective effect to the reversal of IFNGR1 suppression by type I IFNs, the finding here that fGR1 macrophages similarly circumvent IFNγ-driven IFNGR1 down-regulation suggests that the resistance of fGR1 mice may reflect the reversal of this process as well. The ability of IFNγ to dampen myeloid cell IFNGR1 may also be important to ensure dampening of macrophage responsiveness in the context of infections where pathogen burden is insufficiently high to drive a strong type I IFN response (47). Clearly, further studies will be needed to distinguish between the relative contributions of type I versus type II IFN-driven regulation of IFNGR1 in mediating susceptibility to various infections.

Given the importance of IFNγ in mediating inflammation during an innate immune response and the potency of this molecule, conserved mechanisms evolved to control expression of IFNγ itself (48, 49, 50). The work here further shows that modulation of IFNGR1 by IFNγ itself is an evolutionarily conserved mechanism to calibrate macrophage responsiveness to IFNγ. Our data further indicated that the threshold for STAT protein activation is calibrated during macrophage activation by IFNγ through modulation of surface IFNGR1 abundance. This argues that initial exposure of macrophages to IFNγ establishes a checkpoint to ensure full macrophage activation only occurs in the presence of persistent or increasing IFNγ. This likely helps explain why a prolonged exposure to IFNγ is required for full macrophage activation (32) and may have evolved to ensure transient IFNγ production in response to "non-dangerous" PAMPs or DAMPs or a limited translocation of microbes at epithelial barriers does not elicit an overly severe inflammatory response. Transient IFNγ does trigger STAT activation and drive alterations in myeloid cell gene expression, however. This initial IFNγ stimulus corresponds to a "priming" event that can shape the subsequent myeloid cell response. Interestingly, previous studies showed that priming with IFNγ dampened macrophage responses to LPS exposure and exerted anti-inflammatory effects (51). Monocytes primed in vivo by limited IFNγ production were also shown to play a regulatory role in the context of a parasite infection (52). These

**(D)** Luciferase activity from lysates of RAW 264.7 reporter cells with luciferase driven by the proximal *Ifngr1* promoter. Luciferase activity values were normalized to those of the respective untreated cells. Values were derived from three separate experiments using two independently transfected *Ifngr1* promoter-luciferase cell lines. Statistical significance indicates comparison of relative light units between unstimulated ("0 h"; dashed line) and stimulated groups by one-way ANOVA and Dunnett's post hoc test. **(C, E)** WT BMDMs were treated as in (C). ChIP assays were performed for Egr3. Murine primers that amplify an Egr site in the proximal *Ifngr1* promoter were used to quantify immunoprecipitated chromatin by qPCR. **(C, F)** Human THP-1 cells were treated as in (C). ChIP assays were performed for H3K4me3. Human primers that amplify putative enhancer regions upstream of the *Ifngr1* promoter were used to quantify H3K4me3-associated chromatin by qPCR. **(C, E, F)** Graphs depict fold enrichment over isotype values normalized to those of the respective unstimulated cells (relative fold enrichment over isotype = fold enrichment from treated sample/average fold enrichment from untreated sample). **(G)** Naïve WT and fGR1 splenic monocytes were treated for 8 h ± 100 U/ml recombinant IFNβ (blue) or IFNγ (red). Graph depicts relative gMFI IFNGR1 on gated splenic monocytes. For each panel, bar graph represents the mean ± SD of pooled values per condition (n = 3 independent experiments); n.s., *P < 0.05, **P < 0.01, ***P < 0.001 by one-way ANOVA and Dunnett's or Tukey's post-hoc test for comparison between untreated and other groups or comparison between conditions. n.s., not significant.

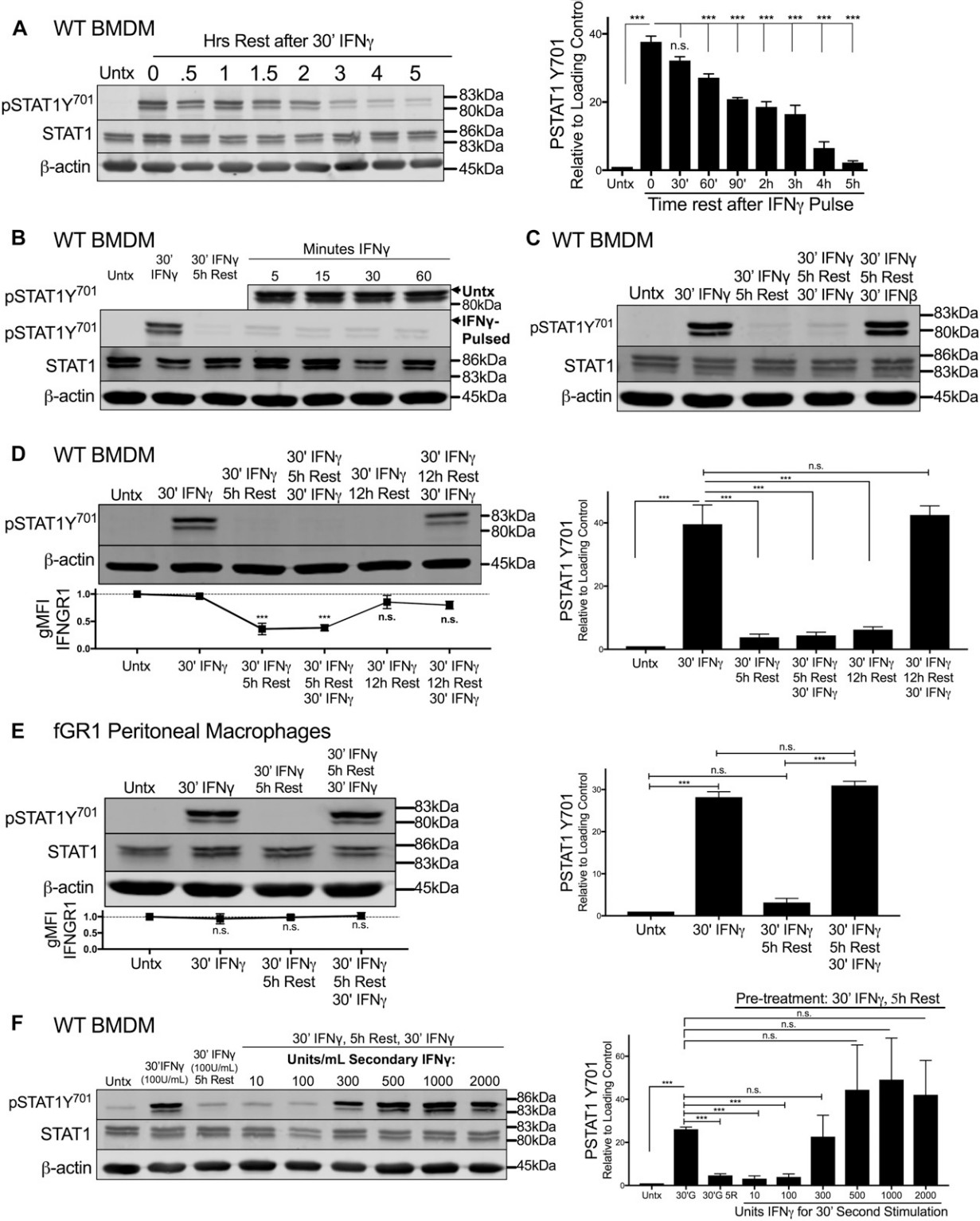

**Figure 4. IFNγ-stimulated myeloid cells become refractory to subsequent IFNγ-induced pSTAT1Y701.**
WT BMDMs or fGR1 peritoneal macrophages were stimulated and lysed. Representative immunoblots depict lysates probed for pSTAT1Y701, Total STAT1, and β-actin. **(A)** WT BMDMs were treated for 30 min with 100 U/ml IFNγ, washed with PBS, and rested for 0, 0.5, 1, 1.5, 2, 3, 4, or 5 h in cytokine-free media. **(B)** Top row depicts WT BMDM lysates probed at 5, 15, 30, or 60 min post stimulation with IFNγ with no pretreatment ("Untx," right side). Second row depicts pSTAT1Y701 from the following treatments: lane 1 (L1), untreated ("Untx"); L2, 30 min (30') IFNγ; L3–L7, pretreated with 30' IFNγ "pulse," PBS wash, 5 h rest in cytokine-free medium followed by secondary stimulation "hit" of 100 U/ml IFNγ for 0, 5, 15, 30, or 60 min. **(C)** WT BMDM lysates probed from the following treatments: lane 1 (L1), untreated ("Untx"); L2, 30 min (30') IFNγ; L3, pretreated with

findings are consistent with the notion that gene expression triggered by a limited IFNγ stimulation may primarily have an inflammation dampening effect.

In summary, our results have expanded mechanistic understanding of how myeloid cell responses to IFNγ are regulated and provide novel insights regarding the impact and likely importance of IFNGR1 down-regulation in the control of myeloid cell activation, tolerance, and susceptibility to infectious and inflammatory diseases. Nevertheless, there are several limitations of our studies. First, although we observed IFNGR1 down-regulation in a widely used in vivo infection model and showed that IFNγ could drive this response in cultured mouse and human cells, it remains to be seen precisely how this regulatory mechanism impacts myeloid cell activation and host resistance in this or other infection models. Second, whereas we showed that IFNγ stimulation alters occupancy of enhancer regions upstream of the *Ifngr1* gene and this correlates with gene silencing, reduced IFNGR1, and impaired responses to subsequent IFNγ, proving of direct cause–effect relationships will require additional experimentation. Future work will also be needed to identify specific targets and strategies to therapeutically manipulate or exploit myeloid cell *Ifngr1* expression towards enhancing or dampening of myeloid cell activation in the context of infectious, cancerous, or inflammatory disease settings.

# Materials and Methods

### Mice

Adult male and female mice were used at 8–12 wk of age. WT C57BL/6 and B6.*Ifngr1*$^{-/-}$ (*Ifngr*$^{-/-}$) mice were from Jackson laboratory and B6.*Ifnar1*$^{-/-}$ (*Ifnar1*$^{-/-}$) mice were described previously ([22]). fGR1 mice were previously described ([18]). All mice were maintained in a specific pathogen-free colony in the University of Colorado Anschutz Medical Campus Office of Laboratory Animal Research.

### Bacterial infections

*L. monocytogenes* (Lm; strain 10403s) was thawed from frozen stocks and grown to log phase (OD600 = 0.1) in tryptic soy broth (MP Biomedicals), supplemented with 50 μg/ml of streptomycin. Lm was diluted in PBS and $10^4$ CFUs were injected to mice i.v. in the lateral tail vein. For cytokine depletion experiments, monoclonal antibodies were diluted in PBS to a concentration of 2.5 mg/ml. Each mouse received 0.5 mg of antibody in 200 μl by intraperitoneal (I.P.)

injection 24 h after infection. IFNγ was depleted using α-IFNγ (XMG1.2; BioXcell). Type I IFNs signaling was blocked using α-IFNAR1 (MAR-1; BioXcell). Spleens were harvested into RP10 media (complete Roswell Park Memorial Institute [RPMI] media [Gibco] supplemented with 10% FBS, 1% sodium pyruvate, 1% L-glutamine, and 1% penicillin/streptomycin) at 0, 24, or 72 hpi then transferred to a digestion solution of 1 mg/ml of collagenase type IV in HBSS plus cations (Gibco). 1 ml of digestion solution was injected into each spleen. After a 25-min incubation at 37°C, 0.5 mM EDTA was added to suppress collagenase activity and spleens were physically disrupted and washed through a 70-μM cell strainer with RPMI + P/S. The cell suspensions were next treated with RBC lysis buffer (0.15 M $NH_4Cl$, 10 mM $KHCO_3$, and 0.1 mM $Na_2EDTA$, pH 7.4) for 3 min, quenched with 10 ml RP10, and centrifuged at 500$g$ for 5 min. Splenocytes were then analyzed by FACS.

### Flow cytometry

Mouse cells were incubated in anti-CD16/32 (2.4G2 hybridoma supernatant) to block Fc receptors then stained by incubation in FACS buffer (1% BSA, 0.01% NaN3, PBS) containing fluorophore-labeled antibodies to mouse proteins that included: anti-CD11b (M1/70; eBioscience), anti-CD11c (N418; BioLegend), anti-Ly6C (Hk1.4; eBioscience), anti-Ly6G (1A8; BioLegend), anti-CD90.2 (53–2.1; eBioscience), anti-IgM (II/4I; eBioscience), and anti-F480 (CL-A3-1; Bio-Rad). Biotinylated anti-IFNGR1/CD119 (2E2; BD Bioscience) and biotinylated anti-IFNGR2 (REA381; Miltenyi Biotec) were stained with secondary streptavidin-APC (eBioscience). For live/dead staining, the cells were stained with LIVE/DEAD Fixable Aqua Dead Cell Stain Kit (#L34965; Thermo Fisher Scientific) before Fc block and surface staining. Monocytes were gated as live singlet cells staining positive for CD11b, F4/80, and Ly6C and negative for Ly6G. For human cells, anti-CD14 (61D3; BD Biosciences), CD3 (OKT3; BD Biosciences), CD19 (HIB19; BD Biosciences), and CD119/IFNGR1 (GIR-208; Thermo Fisher Scientific) were used. After surface staining, the washed cells were fixed in 2–4% paraformaldehyde then analyzed using a FACSCalibur (BD Biosciences) or LSR Fortessa (BD Biosciences). Flow data were processed using FlowJo software (TreeStar).

### Serum cytokine analysis

To quantify IFNγ production, serum was obtained by allowing blood from cardiac puncture to clot for 15 min at room temperature. After centrifugation in polypropylene tubes (Sarstedt), serum was collected and transferred to a new microtube and stored at

---

30′ IFNγ "pulse," PBS wash, 5 h rest in cytokine-free media; L4, as in L3 followed by secondary "hit" of 30′ 100 U/ml IFNγ; L5, as in L3 followed by secondary "hit" of 30′ 100 U/ml IFNβ. **(D)** WT BMDM lysates probed from the following treatments: lane 1 (L1), untreated ("Untx"); L2, 30 min (30′) IFNγ; L3, pretreated with 30′ IFNγ "pulse," PBS wash, 5 h rest in cytokine-free medium; L4, as in L3 followed by secondary "hit" of 30′ 100 U/ml IFNγ; L5, pretreated with 30′ IFNγ "pulse," PBS wash, 12 h rest in cytokine-free medium; L6, as in L5 followed by secondary "hit" of 30′ 100 U/ml IFNγ. **(E)** fGR1 peritoneal macrophages probed from the following treatments: lane 1 (L1), untreated ("Untx"); L2, 30 min (30′) IFNγ; L3, pretreated with 30′ IFNγ "pulse," PBS wash, 5 h rest in cytokine-free medium; L4, as in L3 followed by secondary "hit" of 30′ 100 U/ml IFNγ. **(F)** Top row depicts WT BMDM lysates probed for pSTAT1Y$^{701}$ from the following treatments: lane 1 (L1), untreated ("Untx"); L2, 30 min (30′) 100 U/ml IFNγ; L3, pretreated with 100 U/ml 30′ IFNγ "pulse," PBS wash, 5 h rest in cytokine-free media; L4–L9, as in L3 followed by 30′ secondary "hit" with increasing concentrations of IFNγ: 10, 100, 300, 500, 1,000, or 2,000 U/ml. For each panel, error bars represent SEM. Each bar graph depicts density of pSTAT1Y$^{701}$ bands normalized to β-actin (n = 3 independent experiments). **(D, E)** Each line graph depicts relative gMFI IFNGR1; n.s., ***$P$ < 0.001 by one-way ANOVA and Dunnett's or Tukey's post-hoc test for comparison between untreated and other groups or comparison between conditions. n.s., not significant.
Source data are available for this figure.

−20°C until use. Serum IFNγ was measured using a commercial ELISA (BD Biosciences).

## Cell culture and cytokine stimulations

To obtain BMDMs, bone marrow was flushed from C57BL/6, B6.*Ifngr1*$^{-/-}$, or B6.*Ifnar1*$^{-/-}$ mice femurs and tibias and cultured for 6 d in BM macrophage media (DMEM supplemented with 10% FBS, 1% sodium pyruvate, 1% L-glutamine, 1% penicillin/streptomycin, 2-mercaptoethanol, and 10% L-cell conditioned media). Fresh medium was added at day 3. BMDMs were plated on day 6 for use on day 7 of culture. BMDCs were cultured from C57BL/6, *Ifngr1*$^{-/-}$, or *Ifnar1*$^{-/-}$ mice as previously described (53). RAW 264.7 cells stably transfected with an *ifngr1* promoter-luciferase reporter construct (IFNGR1pr-luc) were previously described (24). IFNGR1pr-luc RAW 264.7 murine macrophage cells were cultured in DM10 media (DMEM supplemented with 10% FBS, 1% sodium pyruvate, 1% L-glutamine, and 1% penicillin/streptomycin). Luciferase activity was measured using a GloMax Microplate Luminometer (Promega). Normalized activity was determined using the formula: (relative luc activity = luc activity from treated sample/average of luc activity from untreated sample). Human monocytic THP-1 cells were cultured in suspension with RP10 media (RPMI 1640 media supplemented with 10% FBS, 1% sodium pyruvate, 1% L-glutamine, and 1% penicillin/streptomycin). Human PBMCs were isolated from de-identified blood donors. Blood was collected in heparin-containing vacuum tubes, and then white blood cells were separated from whole blood as previously described (24). Peritoneal cells were isolated from fGR1 mice as described previously (18). In brief, 10 ml ice cold PBS was used to lavage the peritoneal cavity. Peritoneal cells were plated in DM10 media on tissue culture–treated plates for several hours to enrich for adherence by macrophages, followed by vigorous washes with room temperature PBS. For cytokine stimulations (unless otherwise noted), murine cells were treated with 100 U/ml recombinant mouse IFNγ (#714006; BioLegend) or recombinant mouse IFNβ (#12401-1; PBL), 10 ng/ml recombinant IL-6 (#406-ML-005; R&D Systems), or 50 ng/ml recombinant IL-10 (#14-8101-62; eBioscience). PBMCs and THP-1 cells were treated with 100 U/ml recombinant human IFNγ (#11500-1; PBL) or recombinant human IFNβ (#11410-1; PBL). For inhibition of JAK, WT BMDMs were treated with 5 μg/ml ruxolitinib (#S1378; Selleckchem) for 1 h. To inhibit transcription, the cells were treated with 1 μg/ml actinomycin D (#AC29494-0050; Thermo Fisher Scientific).

## Immunoblotting

At designated time points after cytokine stimulation, total cell lysates from BMDMs or BMDCs were washed three times with room temperature PBS. For experiments including cytokine stimulation followed by rest, BMDMs or BMDCs were treated for 30 min with designated concentration of cytokine, washed once with room temperature PBS, and cultured in cytokine-free media for specified time of rest. The cells were lysed in the culture dish using 0.02% NP-40 supplemented with HALT protease and phosphatase inhibitors (Thermo Fisher Scientific) and 1× SDS–PAGE buffer (0.0625 M Tris-Cl, pH 6.8, 2% SDS, 10% glycerol, 5% 2-ME, and 0.01% bromophenol blue) was added. Equivalent protein amounts were loaded into 10%

acrylamide gels and transferred onto Polyvinylidene difluoride membranes (Millipore). Blots were probed for IFNGR1 (K17), pSTAT1Y$^{701}$ (58D6; Cell Signaling), or Total STAT1 (91-C; Cell Signaling) with β-Actin (8H10D10; Cell signaling) as a loading control on each blot. Blots were developed using the secondary antibodies goat α-rabbit IR 800 (926–32211; LI-COR) and goat α-mouse IR 680 (926–68070; LI-COR) and imaged on an Odyssey CLX (LI-COR). All pSTAT1Y$^{701}$ and IFNGR1 bands were normalized to β-actin on the same blot using ImageStudio ver 4.0 software (LI-COR). Densitometry graphs are pooled from at least three independent pSTAT1Y$^{701}$ or IFNGR1 blots.

## RNA isolation, cDNA synthesis, and qRT-PCR

Total RNA was isolated from splenocytes, BMDMs, or BMDCs in RLT lysis buffer (QIAGEN) and stored at −80°C. The cells were disrupted using a 20-gauge needle and syringe and RNA extracted using the RNeasy Mini Kit following the manufacturer's instructions. cDNA synthesis was conducted using the iScript cDNA Synthesis Kit (Bio-Rad). qRT-PCR was performed using the iTaq Universal SYBR Green Supermix (Bio-Rad). The following exon spanning primer sets were used for murine cells: *Ifng* fw: 5′-AGCTCTTCCTCATGGCTGTT-3′, rev: 5′-TTTTGCCAGTTCCTCCAGAT; *Ifnb* fw: 5′-CATCAACTATAAGCAGCTCCA-3′, rev: 5′-TTCAAGTGGAGAGCAGTTGAG-3′, *Ifna* subtypes fw:5′-CTTCCA-CAGGATCACTGTGTACCT-3′, rev: 5′-TTCTGCTCTGACCACCTCCC-3′; *Gapdh* fw: 5′-ATGGCCTCCAAGGAGTAAG-3′, rev: 5′-CCTAGGCCCCTCCTGTTATT-3′; *Hmbs* fw: 5′-GAGTCTAGATGGCTCAGATAGCATGC-3′, rev: 5′-CCTACA-GACCAGTTAGCGCACATC-3′; *Ifngr1* fw: 5′-AGGTGTATTCGGGTTCCTGG-3′, rev: 5′-AATACGAGGACGGAGAGCTG-3′; *Ifngr2* fw 5′-GTCCTCGCCA-GACTCGTTTT-3′, rev: 5′-CCCGCAGGAAGACTGTGAAT-3′. All qRT-PCRs were conducted in 384-well format with a total reaction volume of 12 μl on a CFX384 Touch Real-Time PCR Detection System (Bio-Rad). The relative transcript abundance values were calculated using the 2$^{-ΔΔCt}$ model with GAPDH and HMBS mRNA as internal controls (54).

## ChIP

The ChIP experiments were performed according to the protocol provided for the Active Motif ChIP Express kit (Active Motif) and described previously (24). Briefly, BMDMs or THP-1 cells were cross-linked with 1% methanol-free formaldehyde for 7 min at room temperature. Fixed cells (7 × 10$^6$ in 300 μl) were resuspended in kit lysis buffer plus protease inhibitors and incubated at 30 min at 4°C. Cell nuclei were pelleted and resuspended in 300 μl of kit shearing buffer plus protease inhibitors. A Covaris S2 sonicator was used to shear the samples using a 27-cycle treatment. 10 μl of supernatant was saved for use as total input DNA. All samples were stored at −80°C until use. Immunoprecipitations were performed overnight at 4°C with protein G magnetic beads (#53033; Active Motif) plus 7 μg of sheared chromatin and antibodies specific for Egr3 (ab75461; Abcam), pS5-RNA pol II (ab5131; Abcam), H3K4me3 (#39915; Active Motif) or control antibodies for mouse IgG (ab46540; Abcam), and human IgG (ab2410; Abcam). After immunoprecipitation, the beads were washed and the immune complexes eluted with kit elution buffer. Reverse cross-linking buffer was added to each eluted supernatant at 1:1. Samples and input DNA were heated for 1 h at 95°C. After treatment with 10 μg/ml proteinase K for 1 h at 37°C, the samples were purified using QIAGEN PCR

purification kit and then used for qPCR. The murine *Ifngr1* promoter primer sequences used to analyze ChIPs were previously described (24), and their sequences were as follows: (pS5-RNA pol II ChIP) fw: 5′-GCAATTGTGTCCCTCGCGCAGGAATGGGCC-3′, rv: 5′-GCTCGTCAAAGCTC-CACTCCCGACC-3′, (Egr3 ChIP) fw: 5′-CCTCAGGCTAGTCCACCCCTTCTCC-3′, rev: 5′-GGAGGCGTGTCTTGGCGGG-3′. Primers used for query of human enhancer region sequences were as follows: (H3K4me3 ChIP) fw: 5′-TGTCTGTCCTTTGAGCGGGA-3′, rev: 5′-CTGTCTCAGCAAGTC-GAGGA-3′; fw: 5′-ATTCAAACCACAGGCTCCGA-3′, rev: 5′-GACTTTGGC-CAAGGCATACCA-3′; fw: 5′-AACTCAAAAGCAAGCGCACA-3′, rev: 5- TCACTCTCAAGCGAACCTGC-3′; fw: 5′-TCTGCTTTATGGAGCGGCTT-3′, rev: 5′-GTGTGCTCGCAAGTGTAACC-3′. qRT-PCR was performed as described above. Graphed results depict fold enrichment over iso-type values normalized to those of the respective untreated cells (relative fold enrichment over isotype = fold enrichment from treated sample/average fold enrichment from the untreated sample).

## Supplementary Information

## Acknowledgements

The authors thank the current and past members of our laboratory for helpful discussion and input into these studies. We especially thank Dr. Sarah Clark, Dr. Daniel McDermott, and Nikki Bortell. Research funding was provided by National Institute of Allergy and Infectious Diseases (NIAID). NIAID grants R33AI102264, R21AI140499, and R01AI131662 (to LL Lenz). WJ Crisler and EM Eshleman received support from T32AI052066.

### Author Contributions

WJ Crisler: conceptualization, data curation, formal analysis, in-vestigation, methodology, and writing—original draft, review, and editing.
EM Eshleman: conceptualization, data curation, formal analysis, and methodology.
LL Lenz: conceptualization, formal analysis, supervision, funding acquisition, project administration, and writing—original draft, review, and editing.

### Conflict of Interest Statement

The authors declare that they have no conflict of interest.

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
