## [Reviewer comments · Life Science Alliance]

Life Science Alliance

Ligand-induced IFNGR1 down regulation calibrates myeloid cell IFN γ responsiveness

William Crisler, Emily Eshleman, and Laurel Lenz
DOI: <https://doi.org/10.26508/lsa.201900447>

Corresponding author(s): Laurel Lenz, University of Colorado School of Medicine

Review Timeline:

Submission Date:	2019-06-04
Editorial Decision:	2019-07-01
Revision Received:	2019-08-26
Editorial Decision:	2019-09-16
Revision Received:	2019-09-23
Accepted:	2019-09-24

Scientific Editor: Andrea Leibfried

Transaction Report:

July 1, 2019

Re: Life Science Alliance manuscript #LSA-2019-00447-T

Dr. Laurel L Lenz
University of Colorado School of Medicine
Immunology and Microbiology
12800 E. 19th Ave
Denver, CO 80045

Dear Dr. Lenz,

Thank you for submitting your manuscript entitled "Ligand-induced IFNGR down regulation calibrates myeloid cell IFN γ responsiveness" to Life Science Alliance. The manuscript was assessed by expert reviewers, whose comments are appended to this letter.

As you will see, your work received somewhat split views from the reviewers. While reviewer #1 and #3 raise some concerns that can get addressed in a revision, reviewer #2 is concerned that the value provided to others remains rather small. This reviewer also notes that the in vivo data (Listeria infection) are problematic, as Listeria infection is always foodborne while you use iv injection, thus questioning the physiological relevance of these data. This reviewer also points out that it has been shown that type I IFN has no detrimental role in foodborne Listeria infection. Reviewer #2 raises some other points that would need experimental revision to clarify inconsistencies (point 1), to add controls for the IFNGR1 abundance measurements (point 2) and to add more support for a transcript level change for IFNGR1 (point 3).

We discussed your work in light of these concerns. Should you think that you'll be able to address the concerns raised by the reviewers, we'd be happy to consider your work for publication here. The issues noted by reviewer #2, however, would need to get addressed in a good way. The concerns of rev#1 and #3 should get addressed, too.

When submitting the revision, please include a letter addressing the reviewers' comments point by

point.

Thank you for this interesting contribution to Life Science Alliance. We are looking forward to receiving your revised manuscript.

Sincerely,

B. MANUSCRIPT ORGANIZATION AND FORMATTING:

Reviewer #1 (Comments to the Authors (Required)):

Title: Ligand-induced IFNGR down regulation calibrates myeloid cell IFN responsiveness

Authors: Crisler, W. Eshleman, E. Lenz, L.

Summary:

Crisler et. al present their work on type I and type II IFNs role in generating a decreased responsiveness of myeloid cells through the mechanism of decreased IFNGR expression.

Recommendation:

General Comments:

- The paper is generally well written, and the authors employ blocking of IFN-gamma, IFNGR1 or both to support their hypothesis that IFN-gamma reduces the abundance of IFNGR1 on myeloid cells.
- One important technical consideration that is not clear: does the antibody to the IFN-gamma receptor detect the receptor when IFN-gamma is bound? If not, then the surface levels as measured by FC may not be accurate.
- Have the authors checked for phosphorylation of STAT3? IFN-gamma does cause this and it would be of interested to know if that also requires higher doses of IFN-gamma in the restimulation experiment.
- There is a significant overlap of data the authors presented here that is already known and published. Please reference Rayamajhi et al. 2010, "Induction of IFN- α enables *Listeria monocytogenes* to suppress macrophage activation by IFN-gamma
- There are several instances of minor grammatical or punctuation errors that should be corrected.

Specific Comments:

- Abstract: The authors briefly mention the impaired recruitment of active pS5-RNAPol2 to the *ifngr1* promoter region. It would be beneficial to include the importance of this in relation to the activation of myeloid cells.
- Introduction: The authors do an excellent job of presenting relevant background information to communicate the importance of IFN-gamma and its respective receptor as well as the role type I IFNs play in the clearance of bacterial infections. Transition sentences between various concepts would improve readability of the manuscript. It is also important to note that the authors describe infection with the bacteria *Listeria monocytogenes*, which induces a type I IFN response. This should be highlighted here.
- Materials and Methods: Please consider separating distinct sections for reader clarification. It is also important to elaborate on particular measurements such as time points, what is stimulated, etc. Describing the gating strategy once within the flow section here should suffice. It would not be required to continually reference how the cells were gated.
- Results/Discussion: The authors do a thorough job of discussing their results: however, the manuscript would be greatly improved through the addition of experiments to further support their hypothesis. It is also important to discuss the limitations of this study, as well as how these results

will be integrated within the already known information. Please also ensure that all figures are referenced throughout the text.

Figures:

oFigure 1: Please consider adding the numerical p value stated within the text, on the graphical depiction. It is of concern that there is a shift seen in the PBS control between Figure 1A and 1B panels. Please clarify the running titles of figures 1A and 1B, what does HPI stand for? Additional cytokine measurements of Type I IFNs would further support the authors claims that IFN-gamma is responsible.

oFigure 2: The authors note that cells were treated with a 1:1 ratio of IFN-gamma to IFN-b, there is concern about the potency of IFN-b in relation to gamma. Further, it would be beneficial to evaluate the impact upon cell vitality. In panel 2D and 2E, it's unclear from the text what GRKO is in relation to the other experimental groups. The data presented in panel 2D seems to be conflicting with the data presented in panel 2F, WT BMDM. Please consider the addition of a treatment control such as PBS to further validate the results.

oFigure 3: It is not evident from the graph which bars are statistically significant in comparison, adding significance bars in addition to the p-values directly within the graph should be considered. In panel 3B, the 60% reduction by IFN-beta is not evident in comparison to IFN-gamma. Also the text refers to 3E as Egr data but the figure shows pS5 pol data and it would seem that 3E is not properly described.

oFigure 4: Please consider adding the statistical significance bars as seen in figure 4C where relevant within the others.

Reviewer #2 (Comments to the Authors (Required)):

The group of L. Lenz previously reported on similar studies in myeloid cells. In mouse model iv infection with *Listeria monocytogenes* leads to rapid loss of responsiveness to IFN γ in myeloid cells (splenic monocytes). This was previously attributed to type I IFN, which is highly induced in this infection model and whose detrimental effect was thus emphasized. To address the mechanism the authors described that type I IFN downregulates the surface level of one IFN γ receptor chain (IFN γ R1) and that this occurs at the transcriptional level via the recruitment of a factor (*egr3*) that silences transcription (Kearney et al J Immunol 2013).

Here the authors present an extension of this work and show that after iv *Listeria* infection downregulation of IFN γ R1 occurs also independently of type I IFN. Moreover, direct stimulation of BMDM with IFN γ leads to a reduction of IFN γ R1 transcripts, and this occurs through a mechanism distinct from what previously described for type I IFN. An alteration of chromatin at the level of the enhancer region of the IFN γ R1 locus is detected by H3K4me3 ChIP data.

The interpretation of these data is that dampening the expression of IFN γ R1, hence of IFN γ responsiveness, may ultimately raise the threshold of STAT1 activation and "calibrate" macrophage function.

While the experiments are for the most part well performed, their biological relevance is uncertain at least in the case of natural *Listeria* infection. The authors should discuss differences between iv injection and foodborne infection. It has been shown that type I IFN has no detrimental role in the natural foodborne infection with *Listeria* (Pitts et al, J Immunol 2016).

Specific points:

- 1) In Figure 1 the surface level of IFN γ R1 is shown to be reduced in splenic monocytes of 72 hr-infected mice and this is attributed to an effect mediated by IFN α /b, as reported. At 24 hr post-infection (Fig. 1B) the reduction of IFN γ R1 is greater but it does not seem to be due to IFN α /b (lack of effect of IFNAR1 Abs injection). The authors suggest that the reduction is caused by IFN γ itself. However IFN γ neutralizing Abs do not appear to prevent the downregulation of the receptor. Why?
- 2) What about the surface level of the other subunit of the IFN γ receptor? I am not totally convinced that what the authors measure is not simply the result of receptor downregulation. Internalization is measured as disappearance of receptor from the cell surface, is a rapid (few minutes) process which (depending on receptor/cell etc) may lead to recycling of the receptor at the cell surface or often to degradation. Thus, abundance of the receptor can be persistently low.
- 3) The finding of reduced IFN γ R1 transcripts is intriguing (Fig. 3A and B) and may need to be strengthened, for instance by measuring IFN γ R2 mRNA levels in the same conditions and by stimulating cells with another cytokine, like IL-6?

Line 232-235: the sentence needs re-writing

Line 283 : Fig. 2C-D should be D,E.

Citation of the different panels of Figure 3 in the text is wrong (Fig. 3A to 3E).

Line 298, no figure of this immunoblot analysis is provided (0, 2, 5, 8hr) (Fig. 2F?)

What is the apparent MW of the IFN γ R1?

Line 338: no statistics is provided in Fig. 3F

Some spelling errors need to be corrected

Overall the work provides a small incremental advance from previous findings.

Reviewer #3 (Comments to the Authors (Required)):

This manuscript submitted by the Lenz group describes the down regulation of ifngr1 in myeloid cells after stimulation by type I and II interferons (IFNs). This study shows the decrease of surface IFN γ Receptor subunit 1 (IFNGR1) in murine immune cells (macrophages and dendritic cells) infected by *Listeria monocytogenes* or stimulated with IFNs beta (type I) and gamma (type II). First, the authors show that the mechanisms by which the two cytokines triggers the decrease of IFNGR1 in these cells are different. Whereas IFN β stimulation leads to the recruitment of the repressive Egr3 protein on ifngr1 promoter region, IFN γ induces the loss of histone methylation on two particular sites of ifngr1 enhancer region which both decrease ifngr1 expression level. Consequently to this loss of IFNGR1 at the cell surface, cells primed with a first pulse of IFN γ cannot be stimulated again by this cytokine before 12h. These data provide some evidence that the control of IFNGR1 level by IFN γ gives the ability to tune cytokine responsiveness in the case of sustained or intensive IFN γ production as during infection.

This study is interesting and well conducted since all the provided data support the claims.

I have only minor comments or modifications to suggest. In particular, the authors need to do some rescue experiment (especially for fig. 4) in which they would transiently express moderate level of exogenous IFNGR1 BMDM not under any ifngr1 promoter/enhancer and see whether STAT1 is still phosphorylated by IFN γ even after priming by a 30 minutes pulse of IFN γ .

The manuscript is intitled: "Ligand-induced IFNGR down regulation calibrates myeloid cell IFN γ responsiveness". It would be better that the authors write IFNGR1 since all their data are on this

subunit without any results on IFNGR2 at the gene or protein level. This modification should be made for the rest of the manuscript (abstract and main text) as well.

Regarding the statistics used in the manuscript, all the figures show unpaired two-tailed t-test (described as "paired" in Statistical Analysis paragraph line 438). In the case of comparison of more than two conditions, one should not use t-test but ANOVA instead.

The Materials and Methods paragraph is missing.

Point-by-point Response to Reviews:

We appreciate the constructive feedback on our manuscript and have incorporated a number of revisions in the current revised version to address suggestions and questions raised by the reviewers. Specifically, we have edited each section of the manuscript to more clearly and accurately emphasize the impact of the studies. We have also included additional data to strengthen conclusions made in the paper and/or clarify various caveats. We are quite pleased with the improvements and hope the editor and reviewers will be satisfied that these revisions merit acceptance of the manuscript.

Reviewer #1 (Comments to the Authors (Required)):

Summary:

Crisler et. al present their work on type I and type II IFNs role in generating a decreased responsiveness of myeloid cells through the mechanism of decreased IFNGR expression.

- The paper is generally well written, and the authors employ blocking of IFN-gamma, IFNGR1 or both to support their hypothesis that IFN-gamma reduces the abundance of IFNGR1 on myeloid cells.

Thank you for these supportive comments and your suggestions below.

- One important technical consideration that is not clear: does the antibody to the IFN-gamma receptor detect the receptor when IFN-gamma is bound? If not, then the surface levels as measured by FC may not be accurate.

This is a critical point that we have included new data to address. These data support the conclusion that the antibody binds to IFNGR1 regardless of receptor ligand occupancy.

Specifically, we show that while ligation of surface IFNGR by IFN γ treatment occurs very rapidly, (elevated STAT1 phosphorylation within 5 min of the treatment in **Fig 4B**) we cannot detect any reduction in cell surface IFNGR1 staining before 2 h after stimulation (**S2G**). Furthermore, when cells are pre-treated with the pan-JAK kinase inhibitor Ruxolitinib to block downstream signaling, the IFN γ treatment failed to reduce IFNGR1 staining even at 8 h after treatment (**Fig S2H**). Finally, immunoblotting demonstrates that total cellular IFNGR1 pools are reduced following the IFN γ treatment. Together, these data (some of which were not included in the prior submission) support the conclusion that binding of IFN γ to cell surface IFNGR1 is not sufficient to reduce detection of surface IFNGR1 by the anti-IFNGR1 and cannot account for the reductions in surface or total cellular IFNGR1 protein.

- Have the authors checked for phosphorylation of STAT3? IFN-gamma does cause this and it would be of interested to know if that also requires higher doses of IFN-gamma in the restimulation experiment.

This is an excellent question that we had not previously addressed. We thus performed new experiments to do so. These revealed a strong phosphorylation of STAT3 (Y⁷⁰⁵) in cells

stimulated with IFN γ (**Fig S2F**). As for pSTAT1Y⁷⁰¹ (**Fig 4, S4**), this pSTAT3Y⁷⁰⁵ signaling decayed to a background level by 5 h after resting cells following a brief pulse with IFN γ and the pSTAT3 signaling was refractory to re-stimulation (**Fig S4C, E**). Thus, both STAT1 and STAT3 signaling are lost when surface IFNGR1 is reduced. Similar to the pSTAT1Y⁷⁰¹ signal, induction of pSTAT3Y⁷⁰⁵ could be restored by increasing the concentration of IFN γ used by 3-fold (**Fig S4E**). We conclude that reductions in IFNGR1 availability following IFN γ stimulation have similar effects on engagement of STAT1 and STAT3 phosphorylation and both STATs can be re-engaged when macrophages are exposed to elevated concentrations of IFN γ .

- There is a significant overlap of data the authors presented here that is already known and published. Please reference Rayamajhi et al. 2010, "Induction of IFN- α enables *Listeria monocytogenes* to suppress macrophage activation by IFN- γ

We recognize there is some overlap with the Rayamajhi *et al* (2010) study as well as with a later Kearney *et al* (2013) paper from our group. We justify this overlap by the fact that we are comparing the effects of IFN γ stimulation (reported here, but not in the previous studies) with previously detailed features of type I IFN-driven down regulation of IFNGR (evaluated in those prior studies). The experiments here using type I IFNs to stimulate IFNGR1 down regulation is necessary as a control to determine distinct features of the IFNGR1 down regulation triggered by IFN β vs. IFN γ .

Still, to reduce the apparent redundancy with the previous studies, we have attempted to further clarify how the current studies diverge from those prior papers in the abstract, introduction, and discussion of the current revised draft. We also streamlined the results to eliminate several phrases and a few figure panels that were more redundant or less essential – favoring instead citation of the previous papers as the reviewer suggests.

Finally, we wish to emphasize for the reviewers that in neither of our prior papers had we investigated the ability of IFN γ stimulation to alter cell surface abundance of IFNGR1. Nor has there been any prior publication of a mechanistic study that demonstrates how IFN γ stimulation reduces myeloid cell surface IFNGR1.

- There are several instances of minor grammatical or punctuation errors that should be corrected.

We have substantially revised the text and in the process have worked to weed out errors of this sort.

Specific Comments:

- Abstract: The authors briefly mention the impaired recruitment of active pS5-RNAPol2 to the *ifngr1* promoter region. It would be beneficial to include the importance of this in relation to the activation of myeloid cells.

The impaired pS5-RNA Pol II recruitment suggests a block in new transcription. However, given this requires further clarification for most readers and is not essential information for capturing the gist of the paper we have removed the reference to pS5-RNA Pol II recruitment from the abstract and expanded the description and discussion of this experiment in the main text.

- Introduction: The authors do an excellent job of presenting relevant background information to communicate the importance of IFN- γ and its respective receptor as well as the role type I

IFNs play in the clearance of bacterial infections. Transition sentences between various concepts would improve readability of the manuscript. It is also important to note that the authors describe infection with the bacteria *Listeria monocytogenes*, which induces a type I IFN response. This should be highlighted here.

We have substantially edited the Introduction. Transitional sentences have been added and we have clarified that type I IFNs are induced during Lm infection as suggested.

- Materials and Methods: Please consider separating distinct sections for reader clarification. It is also important to elaborate on particular measurements such as time points, what is stimulated, etc. Describing the gating strategy once within the flow section here should suffice. It would not be required to continually reference how the cells were gated.

We have worked to correct errors and elaborate on timepoints, etc. in the Materials & Methods section.

- Results/Discussion: The authors do a thorough job of discussing their results; however, the manuscript would be greatly improved through the addition of experiments to further support their hypothesis. It is also important to discuss the limitations of this study, as well as how these results will be integrated within the already known information. Please also ensure that all figures are referenced throughout the text.

We have added a number of experiments to strengthen conclusions of the paper and have attempted to ensure all of the figures are references throughout the text. We have also separated the Results and Discussion.

Data additions include:

- Evaluation of pSTAT3 to show that this readout of IFN γ responsiveness is also affected by IFN γ -induced macrophage refractoriness.
- Evaluation of IFNGR2 surface expression and *Ifngr2* transcript abundance to show the specificity of IFN γ -induced reductions in IFNGR1 and *Ifngr1* (this differs from the type I IFN-stimulated response – which affects both IFNGR1 and IFNGR2).
- Evaluation of myeloid cells from fGR1 mice, which have transgenic expression of IFNGR1 by a macrophage-specific promoter. These cells do not reduce IFNGR1 in response to IFN γ , nor do they experience an IFN γ -induced refractory state as measured by induction of pSTAT1.
- In the treatment conditions that include both IFN β and IFN γ , we performed a titration where one cytokine remained at 100 U/mL and the other was added at either 10, 100, or 1000 U/mL. In these conditions, we also analyzed cells for viability and gated on live cells for expression of IFNGR1.

The discussion now additionally references limitations of this study and includes new text to better integrate our results with regards to previous gaps in knowledge.

Figures:

oFigure 1: Please consider adding the numerical p value stated within the text, on the graphical depiction. It is of concern that there is a shift seen in the PBS control between Figure 1A and 1B panels. Please clarify the running titles of figures 1A and 1B, what does HPI stand for? Additional cytokine measurements of Type I IFNs would further support the authors claims that IFN-gamma is responsible.

We have used bars and asterisks to indicate p-values in all figures.

The shift in PBS seen is due to differences in abundance of type I and II IFNs at 24 and 72 hpi and reflects their additive effects at 24 hpi (thus the trend towards lower staining at this time).

Hpi stands for “hours post infection,” which is now defined in the text in the third sentence of the Results section.

The data reported in **Fig 1E** and **Fig 1F** are novel and demonstrate that there is an induction of *ifnb* and *ifna* subtypes at throughout the first 72 h after systemic *Listeria* infection. While we could use ELISAs to measure these type I IFN proteins, our prior experience has shown that these ELISAs are not very sensitive and even when effects of the type I IFNs are clear it can be difficult to detect the proteins themselves. Further, we believe the results from our *in vitro* and *ex vivo* models clearly show that IFN γ -induces reductions in IFNGR1/*Ifngr1* that are independent of the type I IFNs because they occur in cells from *Ifnar1*^{-/-} mice. For these reasons, we have not further sought to measure type I IFN proteins in the i.v. infection.

oFigure 2: The authors note that cells were treated with a 1:1 ratio of IFN-gamma to IFN-b, there is concern about the potency of IFN-b in relation to gamma. Further, it would be beneficial to evaluate the impact upon cell vitality.

We appreciate the reviewer raising these concerns and have striven to address them with new data. Specifically, we evaluated cell viability in BMDM and showed this was not affected by treatment with 100 U/mL IFN β or IFN γ (**Fig S2I**). However, the combination of IFN β +IFN γ modestly but significantly ($p=0.0159$) reduced percent viable cells (**Fig S2I**). Altering the concentration ratio of the cytokines showed at higher concentrations further reductions in viability. However, when we specifically gated on live cells the gMFI of IFNGR1 was similarly affected regardless if there was higher or less cell death.

In panel 2D and 2E, it's unclear from the text what GRKO is in relation to the other experimental groups. The data presented in panel 2D seems to be conflicting with the data presented in panel 2F, WT BMDM. Please consider the addition of a treatment control such as PBS to further validate the results.

The “GRKO” here referred to BMDM grown from *Ifngr1*^{-/-} mice. To make this clearer, the genotype of origin of the BMDM is now plainly listed in the figure itself. Data in **Fig 2D** and **Fig 2F** (now 2E) were carefully compared and densitometry of replicate blots from 2E are now included in Fig S3A. We did not note any conflict. The Reviewer may have been misled as there remains a high level of IFNGR1 protein in the IFN β -treated *Ifnar1*^{-/-} cells, which is consistent with the lack of response of these cells to the type I IFN stimulation.

oFigure 3: It is not evident from the graph which bars are statistically significant in comparison, adding significance bars in addition to the p-values directly within the graph should be considered. In panel 3B, the 60% reduction by IFN-beta is not evident in comparison to IFN-gamma. Also the text refers to 3E as Egr data but the figure shows pS5 pol data and it would seem that 3E is not properly described.

Throughout the panels, significance is listed using the three-asterisk method, where * represents $p \leq 0.05$, ** represents $p \leq 0.01$, and *** represents $p \leq 0.001$. These definitions of p values represented by are listed in each figure legend. Unless otherwise noted with a

significance bar, the asterisk refers to a comparison of that given set of values to untreated or mock treated (represented by the dashed line at the level of untreated). To reduce confusion, this is now plainly defined in each figure legend. Where appropriate, we have listed the p value in the text, but we consider adding all p values within the graph as redundant considering the inclusion of asterisks and their clear definitions in each figure legend. We also have rearranged the figure panels in Figure 3 to better incorporate new data in the supplementals and have striven to correctly annotate each panel in the text.

oFigure 4: Please consider adding the statistical significance bars as seen in figure 4C where relevant within the others.

Certainly. We have added significance bars where appropriate to make comparisons clearer in **Fig 4** (and also **Fig S4**).

Reviewer #2 (Comments to the Authors (Required)):

The group of L. Lenz previously reported on similar studies in myeloid cells. In mouse model iv infection with *Listeria monocytogenes* leads to rapid loss of responsiveness to IFN γ in myeloid cells (splenic monocytes). This was previously attributed to type I IFN, which is highly induced in this infection model and whose detrimental effect was thus emphasized. To address the mechanism the authors described that type I IFN downregulates the surface level of one IFN γ receptor chain (IFN γ R1) and that this occurs at the transcriptional level via the recruitment of a factor (egr3) that silences transcription (Kearney et al J Immunol 2013).

Here the authors present an extension of this work and show that after iv *Listeria* infection downregulation of IFN γ R1 occurs also independently of type I IFN. Moreover, direct stimulation of BMDM with IFN γ leads to a reduction of IFN γ R1 transcripts, and this occurs through a mechanism distinct from what previously described for type I IFN. An alteration of chromatin at the level of the enhancer region of the IFN γ R1 locus is detected by H3K4me3 ChIP data. The interpretation of these data is that dampening the expression of IFN γ R1, hence of IFN γ responsiveness, may ultimately raise the threshold of STAT1 activation and "calibrate" macrophage function.

While the experiments are for the most part well performed, their biological relevance is uncertain at least in the case of natural *Listeria* infection. The authors should discuss differences between iv injection and foodborne infection. It has been shown that type I IFN has no detrimental role in the natural foodborne infection with *Listeria* (Pitts et al, J Immunol 2016).

Thank you for feedback on our manuscript. Our previous work showed only that type I IFNs elicited down regulation of the IFNGR1 in myeloid cells and detailed mechanisms associated with this. These studies utilized the iv *Listeria* infection model, which has proven to be invaluable as a reproducible model of bacterial infection and induces both type I and II IFN production. We did not mean to imply that this model is or is not physiologically relevant to understanding human listeriosis. It is a useful model just as bone marrow-derived cells are useful in dissecting the mechanisms in these studies. Our main focus here is on reporting on novel findings related to IFN γ and its effects on the expression of its own receptor, IFNGR1. As noted in our response to reviewer #1 we have sought to clarify in the revised version of the

manuscript that this is indeed a novel set of observations and mechanistic studies and that type I IFNs are merely used as a control in our studies.

We have included citation of the Pitts manuscript in the revised Discussion as the authors did show that there is little type I IFN produced during infection with this model of oral Lm infection and yet there was still down regulation of the IFNGR1 on myeloid cells in this context. Our studies are likely relevant in this context as they provide a mechanism for type I IFN-independent downregulation of IFNGR1 in myeloid cells. We note that the Pitts paper had actually suggested a role for IFN γ in driving the down regulation they observed but did not demonstrate this was indeed the case nor provide any additional mechanistic information in this context.

Also, with regards to biological relevance we trust that the reviewer accepts that IFN γ is a vital regulator of macrophage activation and inflammation that must be carefully controlled to prevent damage to the host. Our studies here clearly demonstrate the ability of IFN γ to negatively regulate responses to itself. Given that IFN γ is produced in a wide range of infections, inflammatory responses, and is intentionally provided or induced in therapies the finding that it is actively suppressing myeloid cell responsiveness and mechanistic information on how this occurs seems to us to be quite relevant in a number of biological settings.

Specific points:

1) In Figure 1 the surface level of IFN γ R1 is shown to be reduced in splenic monocytes of 72 hr-infected mice and this is attributed to an effect mediated by IFN α /b, as reported. At 24 hr post-infection (Fig. 1B) the reduction of IFN γ R1 is greater but it does not seem to be due to IFN α /b (lack of effect of IFNAR1 Abs injection). The authors suggest that the reduction is caused by IFN γ itself. However IFN γ neutralizing Abs do not appear to prevent the downregulation of the receptor. Why ?

Allow us to clarify that the conclusion drawn from **Fig 1** is that IFN γ *contributes* to the decrease seen in IFNGR1 at 24 hpi. However, type I IFN is also produced throughout the infection (**Fig 1E-F**). Because both IFN types are present, we had to block responses to both IFNs to fully restore IFNGR1 staining.

We agree with the reviewer that the IFNGR1 staining appears modestly lower at 24 versus 72 hpi. Our interpretation is that this reflects additive effects of both cytokines at 24 hpi. Consistent with this interpretation, IFNGR1 staining at 24 hpi increases slightly with either the anti-IFNAR1 or anti-IFN γ treatment alone. Admittedly, the modest differences here or between 24 and 72 hpi are not significant and thus we did not emphasize them in the text.

2) What about the surface level of the other subunit of the IFN γ receptor ?

Good question. We addressed this by staining for surface IFNGR2. IFN γ did not significantly reduce surface IFNGR2 (**Fig S2E**). This result supports the conclusion that IFN γ selectively targets IFNGR1 and also indicates that cytokine ligation of the receptor is not simply driving internalization of the entire IFNGR complex.

I am not totally convinced that what the authors measure is not simply the result of receptor downregulation. Internalization is measured as disappearance of receptor from the cell surface, is a rapid (few minutes) process which (depending on receptor/cell etc) may lead to recycling of

the receptor at the cell surface or often to degradation. Thus, abundance of the receptor can be persistently low.

We actually refer to this process as down regulation of the IFNGR1 and thus do not argue against use of this nomenclature. We believe the point the reviewer is arguing here is that the mechanism of the down regulation may simply be receptor internalization following ligation.

We likewise at first favored the interpretation that the type I IFN-independent reductions in IFNGR1 were simply due to receptor internalization. However, as noted above, the retention of surface IFNGR2 argue against this being a simple receptor internalization process. Likewise, we show in the revised manuscript that the reduction in surface IFNGR1 requires some 2 hours. Hence, this is not a rapid (few minutes) process. Together with our observations that total cellular IFNGR1 protein is reduced, that transcription of *Ifngr1* is silenced, and that these events are dependent on JAK activity, we believe the argument is compelling that IFN γ signaling modulates occupancy of the *Ifngr1* enhancer to shut down *Ifngr1* transcription. Further support for this model comes from experiments with fGR1 macrophages that are included in the revised manuscript.

3) The finding of reduced of IFN γ R1 transcripts is intriguing (Fig. 3A and B) and may need to be strengthened, for instance by measuring IFN γ R2 mRNA levels in the same conditions and by stimulating cells with another cytokine, like IL-6 ?

We include new data evaluating surface IFNGR2 expression and *Ifngr2* transcript abundance (Fig S2E, S3B). We also include in these figure panels new data stimulating BMDMs with IL-6 or IL-10. We see no significant effects of IFN γ stimulation on *Ifngr2* transcript abundance or surface IFNGR2 and these other cytokines do not impact either IFNGR subunit. Note that blots for pSTAT3 were included to ensure that the concentrations of IL-6 and IL-10 used sufficed to elicit signals in the BMDMs (Fig S2F).

Line 232-235: the sentence needs re-writing

Thank you, we have included substantial revisions/edits to the text in an effort to better clarify the impact and message of the paper.

Line 283 : Fig. 2C-D should be D,E.

Citation of the different panels of Figure 3 in the text is wrong (Fig. 3A to 3E).

We have gone through the paper to carefully check that figure panels are cited correctly. Several new data have been included in the revision. Thus, several of the original panels have been moved to accommodate this.

Line 298, no figure of this immunoblot analysis is provided (0, 2, 5, 8hr) (Fig. 2F ?)
Figure is now included.

What is the apparent MW of the IFN γ R1 ?

The apparent MW of IFNGR1 is ~83kDa. We have added the apparent molecular weight to each immunoblot in this manuscript for completeness.

Line 338: no statistics is provided in Fig. 3F

We have updated the figures to include/improve clarity of statistical analyses.

Some spelling errors need to be corrected
We have thoroughly edited the manuscript.

Overall the work provides a small incremental advance from previous findings.

We appreciate that the reviewer had this opinion after reading the original manuscript and thus went to great lengths to improve the presentation of the data and their impact and novelty. Certainly, the finding that IFN γ reduces responses to itself through this mechanism was surprising to us and many of our colleagues who are quite familiar with our prior work on type I IFN-driven responses have also expressed interest and enthusiasm for the studies.

As we noted above there has not previously been any mechanistic work done to define mechanisms for type I IFN-independent down regulation of the IFNGR1 in myeloid cells. Nor has there been any work done to show that (and how) IFN γ stimulation suppresses or at least raises the threshold for responses to itself through modulation of IFNGR1 abundance. To our knowledge, the mechanism involving IFN γ -stimulated alterations in occupancy at enhancer regions of *Ifngr1* has not been previously described. The fact that both type I and II IFNs (but not IL-6 or IL-10) engage distinct mechanisms to suppress *Ifngr1* transcription in both mice and humans is also novel and striking. As we discuss in the revised manuscript this underscores the importance of this process and likely provides a way to fine tune it. In short, we disagree that the advances here are small and incremental. Rather, we believe them to be substantial and likely to have prolonged impact on our understanding of how macrophage activation by IFN γ is regulated in the context of infections, inflammation, and cancer.

Reviewer #3 (Comments to the Authors (Required)):

This manuscript submitted by the Lenz group describes the down regulation of *ifngr1* in myeloid cells after stimulation by type I and II interferons (IFNs). This study shows the decrease of surface IFN γ Receptor subunit 1 (IFNGR1) in murine immune cells (macrophages and dendritic cells) infected by *Listeria monocytogenes* or stimulated with IFNs beta (type I) and gamma (type II). First, the authors show that the mechanisms by which the two cytokines triggers the decrease of IFNGR1 in these cells are different. Whereas IFN β stimulation leads to the recruitment of the repressive Egr3 protein on *ifngr1* promoter region, IFN γ induces the loss of histone methylation on two particular sites of *ifngr1* enhancer region which both decrease *ifngr1* expression level. Consequently to this loss of IFNGR1 at the cell surface, cells primed with a first pulse of IFN γ cannot be stimulated again by this cytokine before 12h. These data provide some evidence that the control of IFNGR1 level by IFN γ gives the ability to tune cytokine responsiveness in the case of sustained or intensive IFN γ production as during infection. This study is interesting and well conducted since all the provided data support the claims. I have only minor comments or modifications to suggest. In particular, the authors need to do some rescue experiment (especially for fig. 4) in which they would transiently express moderate level of exogenous IFNGR1 BMDM not under any *ifngr1* promoter/enhancer and see whether STAT1 is still phosphorylated by IFN γ even after priming by a 30 minutes pulse of IFN γ .

Excellent suggestion. To address this, we used peritoneal macrophages from fGR1 transgenic mice, in which expression of a functional, flag-tagged IFNGR1 is driven from the macrophage-

specific *c-fms* promoter (Eshleman et al, 2017). Macrophages from fGR1 mice showed increased pSTAT1Y⁷⁰¹ at 30 min after primary IFN γ treatment and lost this pSTAT1 after a 5 h rest (**Fig 4E**). However, the primary IFN γ treatment failed to down regulate IFNGR1 in these cells (determined by surface staining) and pSTAT1 was strongly induced by the secondary IFN γ stimulation (**Fig 4E**). These data indicate that expression of the fGR1 prevented IFN γ -triggered downregulation of IFNGR1 and prevented suppression of pSTAT1 in our “pulse / rest / hit” experiment. These data strengthen the conclusion that reduced surface IFNGR1 is not driven by receptor internalization and can be overcome by expression of the fGR1, which also prevents loss of macrophage IFN γ responsiveness.

The manuscript is intitled: "Ligand-induced IFNGR down regulation calibrates myeloid cell IFN γ responsiveness". It would be better that the authors write IFNGR1 since all their data are on this subunit without any results on IFNGR2 at the gene or protein level. This modification should be made for the rest of the manuscript (abstract and main text) as well.

Agreed. The title has been adjusted as suggested. We also now include data on IFNGR2, showing that neither IFNGR2 surface expression nor *Ifngr2* transcript abundance is reduced by IFN γ stimulation (**Fig S2E, S3B**).

Regarding the statistics used in the manuscript, all the figures show unpaired two-tailed t-test (described as "paired" in Statistical Analysis paragraph line 438). In the case of comparison of more than two conditions, one should not use t-test but ANOVA instead.

The reference to use of the t-test was incorrect in the original version. This has been corrected. Significance was determined by one-way ANOVA followed by either Tukey's multiple comparisons post hoc test, or Dunnett's post hoc test for comparison exclusively between untreated and other groups. A p-value of < 0.05 was considered significant.

The Materials and Methods paragraph is missing.

We have edited the Materials and Methods section to include our new experiments and to ensure this section is complete and accurate.

September 16, 2019

RE: Life Science Alliance Manuscript #LSA-2019-00447-TR

Dr. Laurel L Lenz
University of Colorado School of Medicine
Immunology and Microbiology
12800 E. 19th Ave
Denver, CO 80045

Dear Dr. Lenz,

Thank you for submitting your revised manuscript entitled "Ligand-induced IFNGR1 down regulation calibrates myeloid cell IFN γ responsiveness". As you will see, reviewer #2 re-assessed your work and appreciates the introduced changes, and we would thus be happy to accept your manuscript for publication in Life Science Alliance, pending final revisions to meet our formatting guidelines:

- please check the callout to figure 2I in your manuscript (should be Figure S2I?)
- please provide the source data for Figure 4F and Figure S4E

A. FINAL FILES:

B. MANUSCRIPT ORGANIZATION AND FORMATTING:

Sincerely,

Reviewer #2 (Comments to the Authors (Required)):

The authors have answered my concerns.

The article is considerably improved and the data are in support of the conclusions.

September 24, 2019

RE: Life Science Alliance Manuscript #LSA-2019-00447-TRR

Dr. Laurel L Lenz
University of Colorado School of Medicine
Immunology and Microbiology
12800 E. 19th Ave
Denver, CO 80045

Dear Dr. Lenz,

Thank you for submitting your Research Article entitled "Ligand-induced IFNGR1 down regulation calibrates myeloid cell IFN γ responsiveness". It is a pleasure to let you know that your manuscript is now accepted for publication in Life Science Alliance. Congratulations on this interesting work.

DISTRIBUTION OF MATERIALS:

Again, congratulations on a very nice paper. I hope you found the review process to be constructive and are pleased with how the manuscript was handled editorially. We look forward to future exciting submissions from your lab.

Sincerely,
